# Gold-YOLO: Efficient Object Detector via Gather-and-Distribute Mechanism

**Chengcheng Wang    Wei He    Ying Nie    Jianyuan Guo    Chuanjian Liu**
**Kai Han**[*]    **Yunhe Wang**[*]
Huawei Noah's Ark Lab
{wangchengcheng11,hewei142,ying.nie,jianyuan.guo,liuchuanjian,
kai.han,yunhe.wang}@huawei.com

## Abstract

In the past years, YOLO-series models have emerged as the leading approaches in the area of real-time object detection. Many studies pushed up the baseline to a higher level by modifying the architecture, augmenting data and designing new losses. However, we find previous models still suffer from information fusion problem, although Feature Pyramid Network (FPN) and Path Aggregation Network (PANet) have alleviated this. Therefore, this study provides an advanced Gather-and-Distribute mechanism (GD) mechanism, which is realized with convolution and self-attention operations. This new designed model named as Gold-YOLO, which boosts the multi-scale feature fusion capabilities and achieves an ideal balance between latency and accuracy across all model scales. Additionally, we implement MAE-style pretraining in the YOLO-series for the first time, allowing YOLO-series models could be to benefit from unsupervised pretraining. Gold-YOLO-N attains an outstanding 39.9% AP on the COCO val2017 datasets and 1030 FPS on a T4 GPU, which outperforms the previous SOTA model YOLOv6-3.0-N with similar FPS by +2.4%. The PyTorch code is available at https://github.com/huawei-noah/Efficient-Computing/tree/master/Detection/Gold-YOLO, and the MindSpore code is available at https://gitee.com/mindspore/models/tree/master/research/cv/Gold_YOLO.

## 1 Introduction

Object detection as a fundamental vision task that aims to recognize the categories and locate the positions of objects. It can be widely used in a wide range of applications, such as intelligent security, autonomous driving, robot navigation, and medical diagnosis. High-performance and low-latency object detector is receiving increasing attention for deployment on the edge devices.

Over the past few years, researchers have extensive research on CNN-based detection networks, gradually evolving the object detection framework from two-stage (*e.g.*, Faster RCNN [43] and Mask RCNN [25]) to one-stage (*e.g.*, YOLO [40]), and from anchor-based (*e.g.*, YOLOv3 [42] and YOLOv4 [2]) to anchor-free (*e.g.*, CenterNet [10], FCOS [47] and YOLOX [11]). [12, 7, 17] studied the optimal network structure through NAS for object detection task, and [16, 23, 19] explore another way to improve the performance of the model by distillation. Single-stage detection models, especially YOLO series models, have been widely welcomed in the industry due to their simple structure and balance between speed and accuracy.

Improvement of backbone is also an important research direction in the field of vision. As described in the survey [20], [26, 27, 60, 21, 62] has achieved a balance between precision and speed, while [9, 36, 22, 18] has shown strong performance in precision. These backbones have improved the performance of the original model in different visual tasks, ranging from high-level tasks like object

---

[*]Corresponding Author.

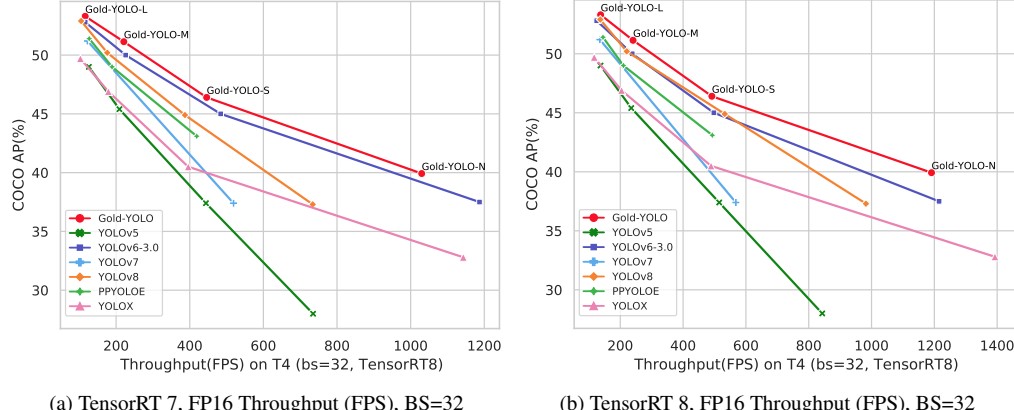

Figure 1: Comparison of state-of-the-art efficient object detectors in Tesla T4 GPU. Both latency and throughput (batch size of 32) are given for a handy reference. (a) and (b) test with TensorRT 7 and 8, respectively.

detection to low-level tasks like image restoration. By using the encoder-decoder structure with the transformer, researchers have constructed a series of DETR-like object detection models, such as DETR [3] and DINO [57]. These models can capture long-range dependency between objects, enabling transformer-based detectors to achieve comparable or superior performance with most refined classical detectors. Despite the notable performance of transformer-based detectors,they fall short when compared to the speed of CNN-based models. Small-scale object detection models based on CNN still dominate the speed-accuracy trade-off, such as YOLOX [11] and YOLOv6-v3 [33, 49, 14].

We focus on the real-time object detection models, especially YOLO series for mobile deployment. Mainstream real-time object detectors consist of three parts: backbone, neck, and head. The backbone architecture has been widely investigated [42, 44, 9, 36] and the head architecture is typically straight forward, consisting of several convolutional or fully-connected layers. The necks in YOLO series usually use Feature Pyramid Network (FPN) and its variants to fuse multi-level features. These neck modules basically follow the architecture shown in Fig. 3. However, the current approach to information fusion has a notable flaw: when there is a need to integrate information across layers (*e.g.*, level-1 and level-3 are fused), the conventional FPN-like structure fails to transmit information without loss, which hinders YOLOs from better information fusion.

Built upon the concept of global information fusion, TopFormer [59] has achieved remarkable results in semantic segmentation tasks. In this paper, we expanding on the foundation of TopFormer's theory, propose a novel Gather-and-Distribute mechanism (GD) for efficient information exchanging in YOLOs by globally fusing multi-level features and injecting the global information into higher levels. This significantly enhances the information fusion capability of the neck without significantly increasing the latency, improving the model's performance across varying object sizes. Specifically, GD mechanism comprises two branches: a shallow gather-and-distribute branch and a deep gather-and-distribute branch, which extract and fuse feature information via a convolution-based block and an attention-based block, respectively. To further facilitate information flow, we introduce a lightweight adjacent-layer fusion module which combines features from neighboring levels on a local scale. Our Gold-YOLO architectures surpasses the existing YOLO series, effectively demonstrating the effectiveness of our proposed approach.

To further improve the accuracy of the model, we also introduce a pre-training method, where we pre-train the backbone on ImageNet 1K using the MAE method, which significantly improves the convergence speed and accuracy of the model. For example, our Gold-YOLO-S with pre-training achieves 46.4% AP, which outperforms the previous SOTA YOLOv6-3.0-S with 45.0% AP at similar speed.

## 2 Related works

### 2.1 Real-time object detectors

After years of development, the YOLO-series model has become popular in the real-time object detection area. YOLOv1-v3 [40, 41, 42] constructs the initial YOLOs, identifies a single-stage

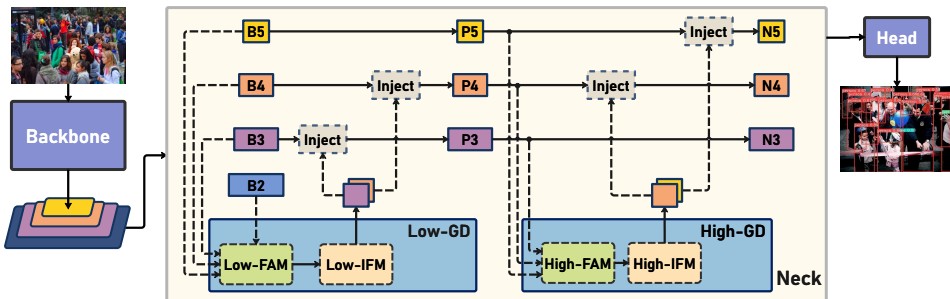

Figure 2: The architecture of the proposed Gold-YOLO.

detection structure consisting of three parts, backbone-neck-head, predicts objects of different sizes through multi-scale branches, become a representative single-stage object detection model. YOLOv4 [2] optimizes the previously used darknet backbone structure and propose a series of improvements, like the Mish activation function, PANet and data augmentation methods. YOLOv5 [13] inheriting the YOLOv4 [2] scheme with improved data augmentation strategy and a greater variety of model variants. YOLOX [11] incorporates Multi positives, Anchor-free, and Decoupled Head into the model structure, setting a new paradigm for YOLO-model design. YOLOv6 [33, 32] brings the reparameterization method to YOLO-series models for the first time, proposing EfficientRep Backbone and Rep-PAN Neck. YOLOv7 [49] focuses on analyzing the effect of gradient paths on the model performance and proposes the E-ELAN structure to enhance the model capability without destroying the original gradient paths. The YOLOv8 [14] takes the strengths of previous YOLO models and integrates them to achieve the SOTA of the current YOLO family.

## 2.2 Transformer-base object detection

Vision Transformer (ViT) emerged as a competitive alternative to convolutional neural networks (CNNs) that are widely used for different image recognition tasks. DETR [3] applies the transformer structure to the object detection task, reconstructing the detection pipeline and eliminating many hand-designed parts and NMS components to simplify the model design and overall process. Combining the sparse sampling capability of deformable convolution with the global relationship modeling capability of transformer, Deformable DETR [63] improve convergence speed while improve model speed and accuracy. DINO [57] first time introduced Contrastive denoising, Mix query selection and a look forward twice scheme. The recent RT-DETR [37] improved the encoder-decoder structure to solve the slow DETR-like model problem, outperforming YOLO-L/X in both accuracy and speed. However, the limitations of the DETR-like structure prevent it from showing sufficient dominance in the small model region, where YOLOs remain the SOTA of accuracy and velocity balance.

## 2.3 Multi-scale features for object detection

Traditionally, features at different levels carry positional information about objects of various sizes. Larger features encompass low-dimensional texture details and positions of smaller objects. In contrast, smaller features contain high-dimensional information and positions of larger objects. The original idea behind Feature Pyramid Networks (FPN) proposed by [35] is that these diverse pieces of information can enhance network performance through mutual assistance. FPN provides an efficient architectural design for fusing multi-scale features through cross-scale connections and information exchange, thereby boosting the detection accuracy of objects of varied sizes.

Based on FPN, the Path Aggregation Network (PANet) [50] incorporates a bottom-up path to make information fusion between different levels more adequate.Similarly, EfficientDet [45] presents a new repeatable module (BiFPN) to increase the efficiency of information fusion between different levels. M2Det [61] introduced an efficient MLFPN architecture with U-shape and Feature Fusion Modules. Ping-Yang Chen [5] improved interaction between deep and shallow layers using bidirectional fusion modules. Unlike these inter-layer works, [38] explored individual feature information using the Centralized Feature Pyramid (CFP) method. Additionally, [54] extended FPN with the Asymptotic Feature Pyramid Network (AFPN) to interact across non-adjacent layers. In response to FPN's limitations in detecting large objects, [31] proposed a refined FPN structure. YOLO-F [6] achieved

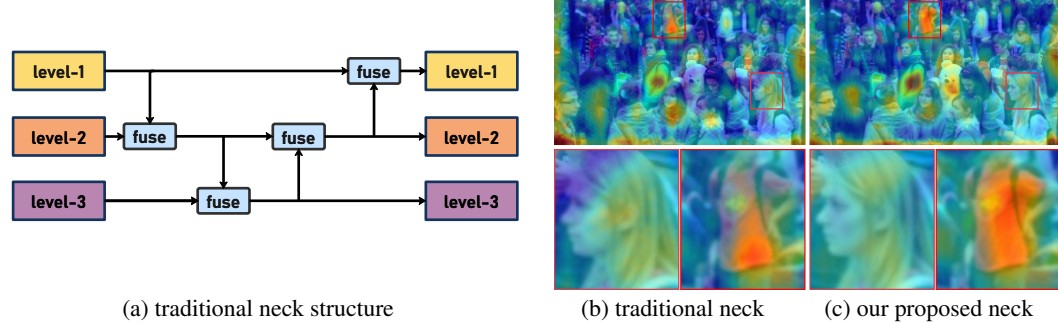

(a) traditional neck structure      (b) traditional neck    (c) our proposed neck

Figure 3: (a) is example diagram of traditional neck information fusion structure. (b) and (c) is AblationCAM [39] visualization

state-of-the-art performance with single-level features. SFNet [34] aligns different level features with semantic flow to improves FPN performance in model. SAFNet [30] introduced Adaptive Feature Fusion and Self-Enhanced Modules. [4] presented a parallel FPN structure for object detection with bi-directional fusion.However, due to the excessive number of paths and indirect interaction methods in the network, the previous FPN-based fusion structures still have drawbacks in low speed, cross-level information exchange and information loss.

However, due to the excessive number of paths and indirect interaction methods in the network, the previous FPN-based fusion structures still have drawbacks in low speed, cross-level information exchange and information loss.

## 3 Method

### 3.1 Preliminaries

The YOLO series neck structure, as depicted in Fig.3, employs a traditional FPN structure, which comprises multiple branches for multi-scale feature fusion. However, it only fully fuse features from neighboring levels, for other layers information it can only be obtained indirectly '*recursively*'. In Fig.3, it shows the information fusion structure of the conventional FPN: where existing level-1, 2, and 3 are arranged from top to bottom. FPN is used for fusion between different levels. There are two distinct scenarios when level-1 get information from the other two levels:

1) If level-1 seeks to utilize information from level-2, it can directly access and fuse this information.

2) If level-1 wants to use level-3 information, level-1 should recursively calling the information fusion module of the adjacent layer. Specifically, the level-2 and level-3 information must be fused first, then level-1 can indirectly obtain level-3 information by combining level-2 information.

This transfer mode can result in a significant loss of information during calculation. Information interactions between layers can only exchange information that is selected by intermediate layers, and not selected information is discarded during transmission. This leads to a situation where information at a certain level can only adequately assist neighboring layers and weaken the assistance provided to other global layers. As a result, the overall effectiveness of the information fusion may be limited.

To avoid information loss in the transmission process of traditional FPN structures, we abandon the original recursive approach and construct a novel gather-and-distribute mechanism (GD). By using a unified module to gather and fuse information from all levels and subsequently distribute it to different levels, we not only avoid the loss of information inherent in the traditional FPN structure but also enhance the neck's partial information fusion capabilities without significantly increasing latency. Our approach thus allows for more effective leveraging of the features extracted by the backbone, and can be easily integrated into any existing backbone-neck-head structure.

In our implementation, the process *gather* and *distribute* correspond to three modules: Feature Alignment Module (FAM), Information Fusion Module (IFM), and Information Injection Module (Inject).

- The *gather* process involves two steps. Firstly, the FAM collects and aligns features from various levels. Secondly, IFM fuses the aligned features to generate global information.
- Upon obtaining the fused global information from the *gather* process, the inject module *distribute* this information across each level and injects it using simple attention operations, subsequently enhancing the branch's detection capability.

To enhance the model's ability to detect objects of varying sizes, we developed two branches: low-stage gather-and-distribute branch (Low-GD) and high-stage gather-and-distribute branch (High-GD). These branches extract and fuse large and small size feature maps, respectively. Further details are provided in Sections 4.1 and 4.2. As shown in Fig. 2, the neck's input comprises the feature maps $B2, B3, B4, B5$ extracted by the backbone, where $B_i \in \mathbb{R}^{N \times C_{Bi} \times R_{Bi}}$. The batch size is denoted by $N$, the channels by $C$, and the dimensions by $R = H \times W$. Moreover, the dimensions of $R_{B2}, R_{B3}, R_{B4}$, and $R_{B5}$ are $R, \frac{1}{2}R, \frac{1}{4}R$, and $\frac{1}{8}R$, respectively.

## 3.2 Low-stage gather-and-distribute branch

In this branch, the output $B2, B3, B4, B5$ features from the backbone are selected for fusion to obtain high resolution features that retain small target information. The structure show in Fig.4(a)

**Low-stage feature alignment module.** In low-stage feature alignment module (Low-FAM), we employ the average pooling (AvgPool) operation to down-sample input features and achieve a unified size. By resizing the features to the smallest feature size of the group ($R_{B4} = \frac{1}{4}R$), we obtain $F_{align}$. The Low-FAM technique ensures efficient aggregation of information while minimizing the computational complexity for subsequent processing through the transformer module.

The target alignment size is chosen based on two conflicting considerations: (1) To retain more low-level information, larger feature sizes are preferable; however, (2) as the feature size increases, the computational latency of subsequent blocks also increases. To control the latency in the neck part, it is necessary to maintain a smaller feature size.

Therefore, we choose the $R_{B4}$ as the target size of feature alignment to achieve a balance between speed and accuracy.

**Low-stage information fusion module.** The low-stage information fusion module (Low-IFM) design comprises multi-layer reparameterized convolutional blocks (RepBlock) and a split operation. Specifically, RepBlock takes $F_{align}$ (channel $= sum(C_{B2}, C_{B3}, C_{B4}, C_{B5})$) as input and produces $F_{fuse}$ (channel $= C_{B4} + C_{B5}$). The middle channel is an adjustable value (*e.g.*, 256) to accommodate varying model sizes. The features generated by the RepBlock are subsequently split in the channel dimension into $F_{inj\_P3}$ and $F_{inj\_P4}$, which are then fused with the different level's feature.

The formula is as follows:

$$F_{\text{align}} = Low\_FAM\left(\left[B2, B3, B4, B5\right]\right), \tag{1}$$

$$F_{\text{fuse}} = RepBlock\left(F_{\text{align}}\right), \tag{2}$$

$$F_{\text{inj\_P3}}, F_{\text{inj\_P4}} = Split(F_{\text{fuse}}). \tag{3}$$

**Information injection module.** In order to inject global information more efficiently into the different levels, we draw inspiration from the segmentation experience [48] and employ attention operations to fuse the information, as illustrated in Fig. 5. Specifically, we input both local information (which refers to the feature of the current level) and global inject information (generated by IFM), denoted as $F_{local}$ and $F_{inj}$, respectively. We use two different Convs with $F_{inj}$ for calculation, resulting in $F_{global\_embed}$ and $F_{act}$. While $F_{local\_embed}$ is calculated with $F_{local}$ using Conv. The fused feature $F_{out}$ is then computed through attention. Due to the size differences between $F_{local}$ and $F_{global}$, we employ average pooling or bilinear interpolation to scale $F_{global\_embed}$ and $F_{act}$ according to the size of $F_{inj}$, ensuring proper alignment. At the end of each attention fusion, we add the RepBlock to further extract and fuse the information.

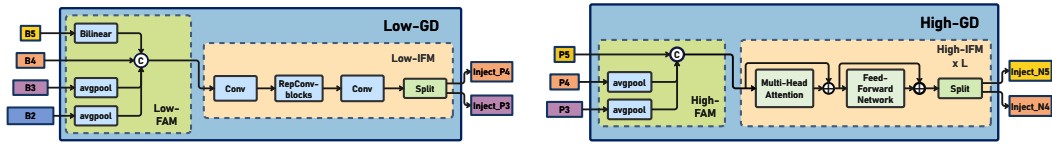

(a) low-stage gather-and-distribute branch  (b) high-stage gather-and-distribute branch

Figure 4: Gather-and-Distribute structure. In (a), the Low-FAM and Low-IFM is low-stage feature alignment module and low-stage information fusion module in low-stage branch, respectively. In (b), the High-FAM and High-IFM is high-stage feature alignment module and high-stage information fusion module, respectively.

In low stage, $F_{local}$ is equal to $Bi$, so the formula is as follows:

$$F_{\text{global\_act\_Pi}} = resize(Sigmoid(Conv_{\text{act}}(F_{\text{inj\_Pi}}))), \tag{4}$$

$$F_{\text{global\_embed\_Pi}} = resize(Conv_{\text{global\_embed\_Pi}}(F_{\text{inj\_Pi}})), \tag{5}$$

$$F_{\text{att\_fuse\_Pi}} = Conv_{\text{local\_embed\_Pi}}(Bi) * F_{\text{ing\_act\_Pi}} + F_{\text{global\_embed\_Pi}}, \tag{6}$$

$$Pi = RepBlock(F_{\text{att\_fuse\_Pi}}). \tag{7}$$

### 3.3 High-stage gather-and-distribute branch

The High-GD fuses the features $\{P3, P4, P5\}$ that are generated by the Low-GD, as shown in Fig.4(b)

**High-stage feature alignment module.** The high-stage feature alignment module (High-FAM) consists of $avgpool$, which is utilized to reduce the dimension of input features to a uniform size. Specifically, when the size of the input feature is $\{R_{P3}, R_{P4}, R_{P5}\}$, $avgpool$ reduces the feature size to the smallest size within the group of features ($R_{P5} = \frac{1}{8}R$). Since the transformer module extracts high-level information, the pooling operation facilitates information aggregation while decreasing the computational requirements for the subsequent step in the Transformer module.

**High-stage information fusion module.** The high-stage information fusion module (High-IFM) comprises the transformer block (explained in greater detail below) and a splitting operation, which involves a three-step process: (1) the $F_{align}$, derived from the High-FAM, are combined using the transformer block to obtain the $F_{fuse}$. (2) The $F_{fuse}$ channel is reduced to $sum(C_{P4}, C_{P5})$ via a $Conv1 \times 1$ operation. (3) The $F_{fuse}$ is partitioned into $F_{inj\_N4}$ and $F_{inj\_N5}$ along the channel dimension through a splitting operation, which is subsequently employed for fusion with the current level feature.

The formula is as follows:

$$F_{\text{align}} = High\_FAM([P3, P4, P5]), \tag{8}$$

$$F_{\text{fuse}} = Transformer(F_{\text{align}}), \tag{9}$$

$$F_{\text{inj\_N4}}, F_{\text{inj\_N5}} = Split(Conv1 \times 1(F_{\text{fuse}})). \tag{10}$$

The transformer fusion module in Eq. 8 comprises several stacked transformers, with the number of transformer blocks denoted by $L$. Each transformer block includes a multi-head attention block, a Feed-Forward Network (FFN), and residual connections. To configure the multi-head attention block, we adopt the same settings as LeViT [15], assigning head dimensions of keys $K$ and queries $Q$ to $D$ (e.g., 16) channels, and $V = 2D$ (e.g., 32) channels. In order to accelerate inference, we substitute the velocity-unfriendly operator, Layer Normalization, with Batch Normalization for each convolution, and replace all GELU activations with ReLU. This minimizes the impact of the transformer module on the model's speed. To establish our Feed-Forward Network, we follow the methodologies presented in [28, 56] for constructing the FFN block. To enhance the local connections of the transformer block, we introduce a depth-wise convolution layer between the two 1x1 convolution layers. We also set the expansion factor of the FFN to 2, aiming to balance speed and computational cost.

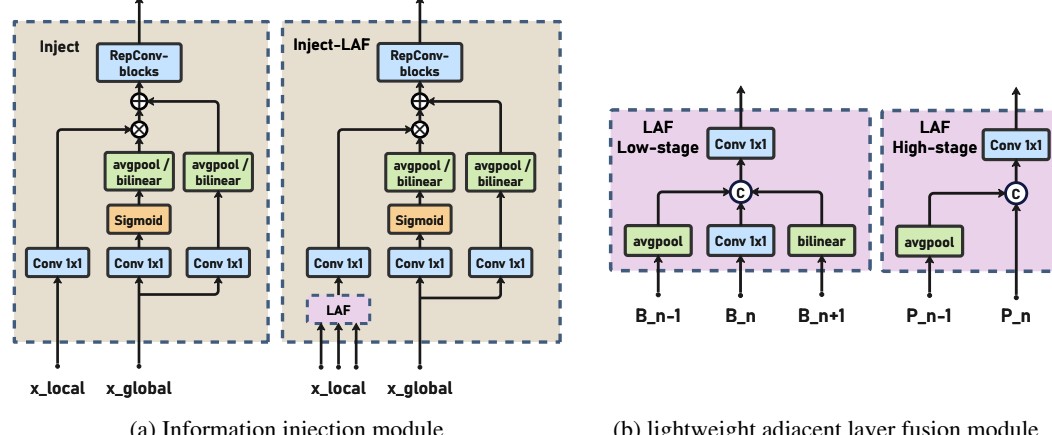

(a) Information injection module
(b) lightweight adjacent layer fusion module

Figure 5: The information injection module and lightweight adjacent layer fusion(LAF) module

**Information injection module.** The information injection module in High-GD is exactly the same as in Low-GD. In high stage, $F_{local}$ is equal to $Pi$, so the formula is as follows:

$$F_{\text{global\_act\_Ni}} = resize(Sigmoid(Conv_{\text{act}}(F_{\text{inj\_Ni}}))), \tag{11}$$

$$F_{\text{global\_embed\_Ni}} = resize(Conv_{\text{global\_embed\_Ni}}(F_{\text{inj\_Ni}})), \tag{12}$$

$$F_{\text{att\_fuse\_Ni}} = Conv_{\text{local\_embed\_Ni}}(Pi) * F_{\text{ing\_act\_Ni}} + F_{\text{global\_embed\_Ni}}, \tag{13}$$

$$Ni = RepBlock(F_{\text{att\_fuse\_Ni}}). \tag{14}$$

### 3.4 Enhanced cross-layer information flow

We have achieved better performance than existing methods using only a global information fusion structure. To further enhance the performance, we drew inspiration from the PAFPN module in YOLOv6 [32] and introduced an Inject-LAF module. This module is an enhancement of the injection module and includes a lightweight adjacent layer fusion (LAF) module that is added to the input position of the injection module.

To achieve a balance between speed and accuracy, we designed two LAF models: LAF low-level model and LAF high-level model, which are respectively used for low-level injection (merging features from adjacent two layers) and high-level injection (merging features from adjacent one layer). There structure is shown in Fig. 5 (b).

To ensure that feature maps from different levels are aligned with the target size, the two LAF models in our implementation utilize only three operators: bilinear interpolation to up-sample features that are too small, average pooling to down-sample features that are too large, and 1x1 convolution to adjust features that differ from the target channel.

The combination of the LAF module with the information injection module in our model effectively balances the between accuracy and speed. By using simplified operations, we are able to increase the number of information flow paths between different levels, resulting in improved performance without significantly increased the latency.

### 3.5 Masked image modeling pre-training

Recent methods such as BEiT [1], MAE [24] and SimMIM [52], have demonstrated the effectiveness of masked image modeling (MIM) for vision tasks. However, these methods are not specifically tailored for convolutional networks (convnets). SparK [46] and ConvNeXt-V2 [51] are pioneers in exploring the potential of masked image modeling for convnets.

In this study, we adopt MIM Pre-training following the SparK's [46] methodology, which successfully identifies and overcomes two key obstacles in extending the success of MAE-style pretraining to

Table 1: Comparisons with other YOLO-series detectors on COCO 2017 val. FPS and latency are measured in FP16-precision on a Tesla T4 in the same environment with TensorRT 7. All our models are trained for 300 epochs. Both the accuracy and the speed performance of our models are evaluated with the input resolution of 640x640. '†' represents that the self-distillation method is utilized, and '⋆' represents that the MIM pre-training method is utilized.

| Method | Input Size | $AP^{val}$ | $AP^{val}_{50}$ | FPS (bs=1) | FPS (bs=32) | Latency (bs=1) | Params | FLOPs |
|---|---|---|---|---|---|---|---|---|
| YOLOv5-N [13] | 640 | 28.0% | 45.7% | 602 | 735 | 1.7 ms | 1.9 M | 4.5 G |
| YOLOv5-S [13] | 640 | 37.4% | 56.8% | 376 | 444 | 2.7 ms | 7.2 M | 16.5 G |
| YOLOv5-M [13] | 640 | 45.4% | 64.1% | 182 | 209 | 5.5 ms | 21.2 M | 49.0 G |
| YOLOv5-L [13] | 640 | 49.0% | 67.3% | 113 | 126 | 8.8 ms | 46.5 M | 109.1 G |
| YOLOX-Tiny [11] | 416 | 32.8% | 50.3% | 717 | 1143 | 1.4 ms | 5.1 M | 6.5 G |
| YOLOX-S [11] | 640 | 40.5% | 59.3% | 333 | 396 | 3.0 ms | 9.0 M | 26.8 G |
| YOLOX-M [11] | 640 | 46.9% | 65.6% | 155 | 179 | 6.4 ms | 25.3 M | 73.8 G |
| YOLOX-L [11] | 640 | 49.7% | 68.0% | 94 | 103 | 10.6 ms | 54.2 M | 155.6 G |
| PPYOLOE-S [53] | 640 | 43.1% | 59.6% | 327 | 419 | 3.1 ms | 7.9 M | 17.4 G |
| PPYOLOE-M [53] | 640 | 49.0% | 65.9% | 152 | 189 | 6.6 ms | 23.4 M | 49.9 G |
| PPYOLOE-L [53] | 640 | 51.4% | 68.6% | 101 | 127 | 10.1 ms | 52.2 M | 110.1 G |
| YOLOv7-Tiny [49] | 416 | 33.3% | 49.9% | 787 | 1196 | 1.3 ms | 6.2 M | 5.8 G |
| YOLOv7-Tiny [49] | 640 | 37.4% | 55.2% | 424 | 519 | 2.4 ms | 6.2 M | 13.7 G |
| YOLOv7 [49] | 640 | 51.2% | 69.7% | 110 | 122 | 9.0 ms | 36.9 M | 104.7 G |
| YOLOv7-E6E [49] | 1280 | 56.8% | 74.4% | 16 | 17 | 59.6 ms | 151.7 M | 843.2 G |
| YOLOv8-N [14] | 640 | 37.3% | 52.6% | 561 | 734 | 1.8 ms | 3.2 M | 8.7 G |
| YOLOv8-S [14] | 640 | 44.9% | 61.8% | 311 | 387 | 3.2 ms | 11.2 M | 28.6 G |
| YOLOv8-M [14] | 640 | 50.2% | 67.2% | 143 | 176 | 7.0 ms | 25.9 M | 78.9 G |
| YOLOv8-L [14] | 640 | 52.9% | 69.8% | 91 | 105 | 11.0 ms | 43.7 M | 165.2 G |
| YOLOv6-3.0-N [32] | 640 | 37.0% / 37.5%† | 52.7% / 53.1%† | 779 | 1187 | 1.3 m s | 4.7 M | 11.4 G |
| YOLOv6-3.0-S [32] | 640 | 44.3% / 45.0%† | 61.2% / 61.8%† | 339 | 484 | 2.9 ms | 18.5 M | 45.3 G |
| YOLOv6-3.0-M [32] | 640 | 49.1% / 50.0%† | 66.1% / 66.9%† | 175 | 226 | 5.7 ms | 34.9 M | 85.8 G |
| YOLOv6-3.0-L [32] | 640 | 51.8% / 52.8%† | 69.2% / 70.3%† | 98 | 116 | 10.3 ms | 59.6 M | 150.7 G |
| Gold-YOLO-N | 640 | 39.6% / 39.9%† | 55.7% / 55.9%† | 563 | 1030 | 1.7 ms | 5.6 M | 12.1 G |
| Gold-YOLO-S | 640 | 45.4% / 46.1%† | 62.5% / 63.3%† | 286 | 446 | 3.3 ms | 21.5 M | 46.0 G |
| Gold-YOLO-M | 640 | 49.8% / 50.9%† | 67.0% / 68.2%† | 152 | 220 | 6.4 ms | 41.3 M | 87.5 G |
| Gold-YOLO-L | 640 | 51.8% / 53.2%† | 68.9% / 70.5%† | 88 | 116 | 11.1 ms | 75.1 M | 151.7 G |
| Gold-YOLO-S⋆ | 640 | 45.5% / 46.4%† | 62.2% / 63.4%† | 286 | 446 | 3.3 ms | 21.5 M | 46.0 G |
| Gold-YOLO-M⋆ | 640 | 50.2% / 51.1%† | 67.5% / 68.5%† | 152 | 220 | 6.4 ms | 41.3 M | 87.5 G |
| Gold-YOLO-L⋆ | 640 | 52.3% / 53.3%† | 69.6% / 70.9%† | 88 | 116 | 11.1 ms | 75.1 M | 151.7 G |

convolutional networks (convnets). These challenges include the convolutional operations' inability to handle irregular and randomly masked input images, as well as the inconsistency between the single-scale nature of BERT pretraining and the hierarchical structure of convnets.

To address the first issue, unmasked pixels are treated as sparse voxels of 3D point clouds and employ sparse convolution for encoding. For the latter issue, a hierarchical decoder is developed to reconstruct images from multi-scale encoded features. The framework adopts a UNet-style architecture to decode multi-scale sparse feature maps, where all spatial positions are filled with embedded masks. We pretrain our model's backbone on ImageNet 1K for multiple Gold-YOLO models, and results in notable improvements

## 4  Experiment

### 4.1  Setups

**Datasets.**   We perform extensive experiments on the Microsoft COCO datasets to validate the proposed detector. For the ablation study, we train on COCO train2017 and validate on COCO val2017 datasets. We use the standard COCO AP metric with a single scale image as input, and report the standard mean average precision (AP) result under different IoU thresholds and object scales.

**Implementation details.**   We followed the setup of YOLOv6-3.0 [32] use the same structure (except for neck) and training configurations. The backbone of the network was implemented with

the EfficientRep Backbone, while the head utilized the Efficient Decoupled Head. The optimizer learning schedule and other setting also same as YOLOv6, *i.e.* stochastic gradient descent (SGD) with momentum and cosine decay on learning rate. Warm-up, grouped weight decay strategy and the exponential moving average (EMA) are utilized. Self-distillation and anchor-aided training (AAT) also be used in training. The strong data augmentations we adopt Mosaic [2, 13] and Mixup [58].

We conducted MIM unsupervised pretraining on the backbone using the 1.28 million ImageNet-1K datasets [8]. Following the experiment settings in Spark [46], we employed a LAMB optimizer [55] and cosine-annealing learning rate strategy, with a masking ratio of 60 % and a mask patch size of 32. For the Gold-YOLO-L models, we employed a batch size of 1024, while for the Gold-YOLO-M models, a batch size of 1152 was used. MIM pretraining was not employed for Gold-YOLO-N due to the limited capacity of its small backbone.

All our models are trained on 8 NVIDIA A100 GPUs, and the speed performance is measured on an NVIDIA Tesla T4 GPU with TensorRT.

## 4.2 Comparisons

Our focus is primarily on evaluating the speed performance of our models after deployment. Specifically, we measure throughput (frames per second at a batch size of 1 or 32) and GPU latency, rather than FLOPs or the number of parameters. To compare our Gold-YOLO with other state-of-the-art detectors in the YOLO series, such as YOLOv5 [13], YOLOX [11], PPYOLOE [53], YOLOv7 [49], YOLOv8 [14] and YOLOv6-3.0 [32], we test the speed performance of all official models with FP16-precision on the same Tesla T4 GPU with TensorRT.

Gold-YOLO-N demonstrates notable advancements, achieving an improvement of 2.6%/2.4%/6.6% compared to YOLOv8-N, YOLOv6-3.0-N, and YOLOv7-Tiny (input size=416), respectively, while offering comparable or superior performance in terms of throughput and latency. When compared to YOLOX-S and PPYOLOE-S, Gold-YOLO-S demonstrates a notable increase in AP by 5.9%/3.1%, while operating at a faster speed of 50/27 FPS (with a batch size of 32).

Gold-YOLO-M outperforms YOLOv6-3.0-M, YOLOX-M and PPYOLOE-M by achieving 1.1%, 4.2% and 2.1% higher AP with a comparable speed. Additionally, it achieves 5.7% and 0.9% higher AP than YOLOv5-M and YOLOv8-M, respectively, while achieving a higher speed. Gold-YOLO-M outperforms YOLOv7 with a significant improvement of 98FPS (batch size = 32), while maintaining the same AP. Gold-YOLO-L also achieves a higher accuracy compared to YOLOv8-L and YOLOv6-3.0-L, with a noticeable accuracy advantage of 0.4% and 0.5% respectively, while maintaining similar FPS at a batch size of 32.

## 4.3 Ablation study

### 4.3.1 Ablation study on GD structure

To verify the validity of our analysis concerning the FPN and to assess the efficacy of the proposed gather-and-distribute mechanism, we examined each module in GD independently, focusing on AP, number of parameters, and latency on the T4 GPU. The Low-GD predominantly targets small and medium-sized objects, whereas the High-GD primarily detect large-sized objects, and the LAF module bolsters both branches. The experimental results are displayed in Table 2 .

Table 2: Ablation study on GD structure. The test model is Gold-YOLO-S on T4 GPU evaluate.

| Low-GD | High-GD | LAF | AP | AP-small | AP-medium | AP-large | FPS $_{(bs=32)}$ | Params | FLOPs |
|---|---|---|---|---|---|---|---|---|---|
| ✓ | | | 44.65% | 25.13% | 49.70% | 60.36% | 454.4 | 18.7 M | 45.1 G |
| | ✓ | | 42.27% | 20.79% | 47.74% | 61.09% | 493.7 | 20.9 M | 43.6 G |
| | | ✓ | 44.36% | 25.04% | 49.53% | 60.64% | 526.2 | 18.1 M | 43.3 G |
| ✓ | ✓ | | 45.57% | 24.90% | 50.38% | 63.50% | 461.8 | 21.5 M | 45.8 G |
| ✓ | ✓ | ✓ | 46.11% | 25.22% | 51.23% | 63.42% | 446.2 | 21.5 M | 46.0 G |

### 4.3.2 Ablation study on LAF

In this ablation study, we conducted experiments to compare the effects of different module designs within the LAF framework and evaluate the influence of varying model sizes on accuracy. The

results of our study provide evidence to support the assertion that the existing LAF structure is indeed optimal. The difference between model-1 and model-2 is whether LAF uses add or concat, and model-3 increase the model size basis of model-2. The model-4 is based on model-3 but discards LAF. The experimental results are displayed in Table 3 .

Table 3: Ablation study on LAF. Use TensorRT 7 on T4 GPU evaluate.

| model | concat | add | increase model size | AP | AP-small | AP-medium | AP-large | FPS (bs=32) | Params | FLOPs |
|---|---|---|---|---|---|---|---|---|---|---|
| 1 | ✓ | | | 46.11% | 25.22% | 51.23% | 63.42% | 446.2 | 21.5 M | 46.0 G |
| 2 | | ✓ | | 45.43% | 24.98% | 50.66% | 62.10% | 413.2 | 20.9 M | 47.5 G |
| 3 | | ✓ | ✓ | 46.49% | 26.32% | 51.29% | 63.43% | 356.0 | 30.8 M | 54.1 G |
| 4 | | | ✓ | 46.47% | 25.27% | 50.80% | 64.25% | 373.8 | 26.4 M | 52.9 G |

#### 4.3.3 Ablation study on other model and task

The GD mechanism is a general concept and can be applied beyond YOLOs. We have extend GD mechanism to other models and obtain significant improvement.

On Instance Segmentation task, we replace different necks in Mask R-CNN and train/test on the COCO instance datasets. The result as shown in the Table 4.

Table 4: Ablation study on Instance Segmentation Task.

| Model | Neck | FPS | Bbox mAP | Bbox mAP:50 | Segm mAP | Segm mAP:50 |
|---|---|---|---|---|---|---|
| MaskRCNN-ResNet50 | FPN | 21.6 | 38.2% | 58.8% | 34.7% | 55.7% |
| MaskRCNN-ResNet50 | AFPN | 19.1 | 36.0% | 53.6% | 31.8% | 50.7% |
| MaskRCNN-ResNet50 | PAFPN | 20.2 | 37.9% | 58.6% | 34.5% | 55.3% |
| MaskRCNN-ResNet50 | GD | 18.7 | 40.7% | 59.5% | 36.0% | 56.4% |

On Semantic Segmentation task, we replace different necks in PointRend and train/test on the Cityscapes datasets. The result as shown in the Table 5.

Table 5: Ablation study on Semantic Segmentation Task.

| Model | Neck | FPS | mIoU | mAcc | aAcc |
|---|---|---|---|---|---|
| PointRend-ResNet50 | FPN | 11.21 | 76.47 | 84.05 | 95.96 |
| PointRend-ResNet50 | GD | 11.07 | 78.54 | 85.60 | 96.12 |
| PointRend-ResNet101 | FPN | 8.76 | 78.30 | 85.71 | 96.23 |
| PointRend-ResNet101 | GD | 7.54 | 80.01 | 86.15 | 96.34 |

Table 6: Performance of GD mechanism on other object detection models.

| model | Neck | FPS | AP |
|---|---|---|---|
| EfficientDet | BiFPN | 6.0 | 34.4 |
| EfficientDet | GD | 5.7 | 38.8 |

On object detection task, we replace different necks in EfficientDet and train/test on the COCO datasets. The result as shown in the Table 6.

## 5 Conclusion

In this paper, we revisit the traditional Feature Pyramid Network (FPN) architecture and critically analyze its constraints in terms of information transmission. Following this, we subsequently developed the Gold-YOLO series models for object detection tasks, achieving state-of-the-art results. In Gold-YOLO we introduce an innovative gather-and-distribute mechanism, strategically designed to enhance the efficacy and efficiency of information fusion and transmission, avoid unnecessary losses, thereby significantly improving the model's detection capabilities. We truly hope that our work will prove valuable in addressing real-world problems and may also ignite fresh ideas for researchers in this field.

# Acknowledgement

We gratefully acknowledge the support of MindSpore [29], CANN (Compute Architecture for Neural Networks) and Ascend AI Processor used for this research.

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

# A    Additional experiment

## A.1    More detailed accuracy and speed data for Gold-YOLO

In this section, we report the test performace of our Gold-YOLO with or without LAF module and pre-training. FPS and latency are measured in FP16-precision on a Tesla T4 in the same environment with TensorRT 7. Both the accuracy and the speed performance of our models are evaluated with the input resolution of 640x640. The result shown in Table 7 .

Table 7: Test results of Gold-YOLO series model on COCO 2017 val. '†' represents that the self-distillation method is utilized, '⋄' represents that the model don't have LAF module, and '⋆' represents that the MIM pre-training method is utilized.

| Method | $\mathbf{AP}^{val}$ | $\mathbf{AP}^{val}_{50}$ | $\mathbf{AP}^{val}_{small}$ | $\mathbf{AP}^{val}_{medium}$ | $\mathbf{AP}^{val}_{large}$ | FPS $_{(bs=32)}$ | Params | FLOPs |
|---|---|---|---|---|---|---|---|---|
| Gold-YOLO-N⋄ | 38.37% / 38.82%† | 54.43% / 54.96%† | 18.37% / 18.40%† | 42.62% / 43.45%† | 55.13% / 56.00%† | 1087 | 5.6 M | 12.0 G |
| Gold-YOLO-S⋄ | 44.47% / 45.57%† | 61.55% / 62.92%† | 24.04% / 24.90%† | 49.34% / 50.38%† | 61.95% / 63.50%† | 462 | 21.5 M | 45.8 G |
| Gold-YOLO-M⋄ | 49.41% / 50.26%† | 66.64% / 67.58%† | 31.19% / 31.59%† | 54.30% / 55.26%† | 65.91% / 67.62%† | 229 | 41.3 M | 86.8 G |
| Gold-YOLO-L⋄ | 51.68% / 52.65%† | 69.07% / 70.25%† | 34.86% / 34.20%† | 56.92% / 57.70%† | 69.00% / 69.71%† | 119 | 75.0 M | 150.6 G |
| Gold-YOLO-N | 39.57% / 39.92%† | 55.70% / 55.94%† | 19.67% / 19.15%† | 44.08% / 44.32%† | 56.98% / 57.75%† | 1030 | 5.6 M | 12.1 G |
| Gold-YOLO-S | 45.36% / 46.11%† | 62.48% / 63.33%† | 25.32% / 25.22%† | 50.21% / 51.23%† | 62.63% / 63.42%† | 446 | 21.5 M | 46.0 G |
| Gold-YOLO-M | 49.77% / 50.86%† | 67.01% / 68.23%† | 32.32% / 31.01%† | 55.29% / 56.24%† | 66.27% / 67.83%† | 220 | 41.3 M | 87.5 G |
| Gold-YOLO-L | 51.84% / 53.16%† | 68.94% / 70.49%† | 34.12% / 34.53%† | 57.36% / 58.60%† | 68.17% / 70.07%† | 116 | 75.1 M | 151.7 G |
| Gold-YOLO-S⋆ | 45.52% / 46.36%† | 62.20% / 63.36%† | 24.66% / 25.26%† | 50.76% / 51.30%† | 63.24% / 63.64%† | 446 | 21.5 M | 46.0 G |
| Gold-YOLO-M⋆ | 50.16% / 51.14%† | 67.52% / 68.53%† | 30.52% / 32.33%† | 55.54% / 56.10%† | 67.64% / 68.55%† | 220 | 41.3 M | 87.5 G |
| Gold-YOLO-L⋆ | 52.25% / 53.28%† | 69.61% / 70.93%† | 33.09% / 33.83%† | 57.77% / 58.92%† | 69.01% / 69.92%† | 116 | 75.1 M | 151.7 G |

## A.2    MIM pre-training ablation experiment

We also compared the Gold-YOLO-S on COCO 2017 validation results for different MIM pre-training epochs without self-distillation. The result shown in Table 8 .

Table 8: Test results on COCO 2017 val for different pre-training epoch setting.

| Epoch | $\mathbf{AP}^{val}$ | $\mathbf{AP}^{val}_{50}$ | $\mathbf{AP}^{val}_{small}$ | $\mathbf{AP}^{val}_{medium}$ | $\mathbf{AP}^{val}_{large}$ |
|---|---|---|---|---|---|
| 400 | 45.39% | 62.17% | 25.01% | 50.28% | 62.74% |
| 600 | 45.48% | 62.18% | 25.56% | 50.63% | 62.85% |
| 800 | 45.52% | 62.20% | 24.66% | 50.76% | 63.24% |

# B    Comprehensive Latency and Throughput Benchmark

## B.1    Model Latency and Throughput on T4 GPU with TensorRT 8

Comparisons with other YOLO-series detectors on COCO 2017 val. FPS and latency are measured in FP16-precision on Tesla T4 in the same environment with TensorRT 8.2. The result shown in Table 9.

Table 9: Comparison of Latency and Throughput in YOLO series model on a T4 GPU using TensorRT 8.2.

| Method | Input Size | FPS (bs=1) | FPS (bs=32) | Latency (bs=1) |
|---|---|---|---|---|
| YOLOv5-N | 640 | 702 | 843 | 1.4 ms |
| YOLOv5-S | 640 | 433 | 515 | 2.3 ms |
| YOLOv5-M | 640 | 202 | 235 | 4.9 ms |
| YOLOv5-L | 640 | 126 | 137 | 7.9 ms |
| YOLOX-Tiny | 416 | 766 | 1393 | 1.3 ms |
| YOLOX-S | 640 | 313 | 489 | 2.6 ms |
| YOLOX-M | 640 | 159 | 204 | 5.3 ms |
| YOLOX-L | 640 | 104 | 117 | 9.0 ms |
| PPYOLOE-S | 640 | 357 | 493 | 2.8 ms |
| PPYOLOE-M | 640 | 163 | 210 | 6.1 ms |
| PPYOLOE-L | 640 | 110 | 145 | 9.1 ms |
| YOLOv7-Tiny | 640 | 464 | 568 | 2.1 ms |
| YOLOv7 | 640 | 128 | 135 | 7.6 ms |
| YOLOv6-3.0-N | 640 | 785 | 1215 | 1.3 m s |
| YOLOv6-3.0-S | 640 | 345 | 498 | 2.9 ms |
| YOLOv6-3.0-M | 640 | 178 | 238 | 5.6 ms |
| YOLOv6-3.0-L | 640 | 105 | 125 | 9.5 ms |
| Gold-YOLO-N | 640 | 657 | 1191 | 1.4 ms |
| Gold-YOLO-S | 640 | 308 | 492 | 3.1 ms |
| Gold-YOLO-M | 640 | 157 | 241 | 6.1 ms |
| Gold-YOLO-L | 640 | 94 | 137 | 10.3 ms |

## B.2 Model Latency and Throughput on V100 GPU with TensorRT 7

Comparisons with other YOLO-series detectors on COCO 2017 val. FPS and latency are measured in FP16-precision on Tesla V100 in the same environment with TensorRT 7.2. The result shown in Table 10 .

Table 10: Comparison of Latency and Throughput in YOLO series model on a V100 GPU using TensorRT 7.2.

| Method | Input Size | FPS (bs=1) | FPS (bs=32) | Latency (bs=1) |
|---|---|---|---|---|
| YOLOv5-N | 640 | 577 | 1727 | 1.4 ms |
| YOLOv5-S | 640 | 449 | 1249 | 1.7 ms |
| YOLOv5-M | 640 | 271 | 698 | 3.0 ms |
| YOLOv5-L | 640 | 178 | 440 | 4.7 ms |
| YOLOX-Tiny | 416 | 569 | 2883 | 1.4 ms |
| YOLOX-S | 640 | 386 | 1206 | 2.0 ms |
| YOLOX-M | 640 | 245 | 600 | 3.4 ms |
| YOLOX-L | 640 | 149 | 361 | 5.6 ms |
| PPYOLOE-S | 640 | 322 | 1050 | 2.4 ms |
| PPYOLOE-M | 640 | 222 | 566 | 4.0 ms |
| PPYOLOE-L | 640 | 153 | 406 | 5.5 ms |
| YOLOv7-Tiny | 640 | 453 | 1565 | 1.7 ms |
| YOLOv7 | 640 | 182 | 412 | 4.6 ms |
| YOLOv6-3.0-N | 640 | 646 | 2660 | 1.2 m s |
| YOLOv6-3.0-S | 640 | 399 | 1330 | 2.0 ms |
| YOLOv6-3.0-M | 640 | 203 | 676 | 4.4 ms |
| YOLOv6-3.0-L | 640 | 125 | 385 | 6.8 ms |
| Gold-YOLO-N | 640 | 574 | 2457 | 1.7 ms |
| Gold-YOLO-S | 640 | 391 | 1205 | 2.5 ms |
| Gold-YOLO-M | 640 | 238 | 633 | 4.0 ms |
| Gold-YOLO-L | 640 | 146 | 365 | 6.6 ms |

## C  Broader impacts and limitations

**Broader impacts.**  The YOLO model can be widely applied in fields such as healthcare and intelligent transportation. In the healthcare domain, the YOLO series models can improve the early diagnosis rates of certain diseases and reduce the cost of initial diagnosis, thereby saving more lives. In the field of intelligent transportation, the YOLO model can assist in autonomous driving of vehicles, enhancing traffic safety and efficiency. However, there are also risks associated with the military application of the YOLO model, such as target recognition for drones and assisting military reconnaissance. We will make every effort to prevent the use of our model for military purposes.

**Limitations.**  Generally, making finer adjustments on the structure will help further improve the model's performance, but this requires a significant amount of computational resources. Additionally, due to our algorithm's heavy usage of attention operations, it may not be as friendly to some earlier hardware support.

## D  CAM visualization

Below are the CAM visualization results of the neck for YOLOv5, YOLOv6, YOLOv7, YOLOv8, and our Gold-YOLO, shown in Fig. 6. It can be observed that our model assigns higher weights to the detected regions of the targets.

And we compare the neck CAM visualization between Gold-YOLO and YOLOv6, shown in Fig. 7.

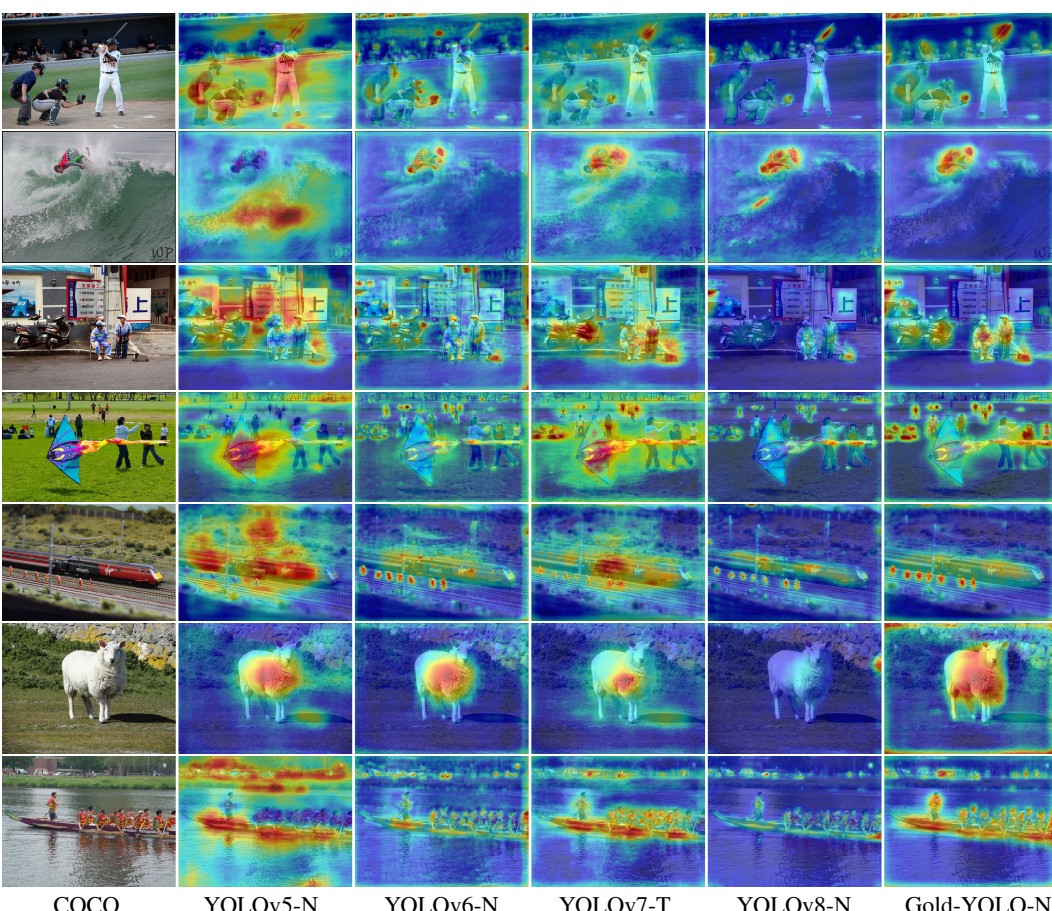

| COCO | YOLOv5-N | YOLOv6-N | YOLOv7-T | YOLOv8-N | Gold-YOLO-N |

Figure 6: The CAM visualization results of the neck for YOLOv5, YOLOv6, YOLOv7, YOLOv8, and our Gold-YOLO.

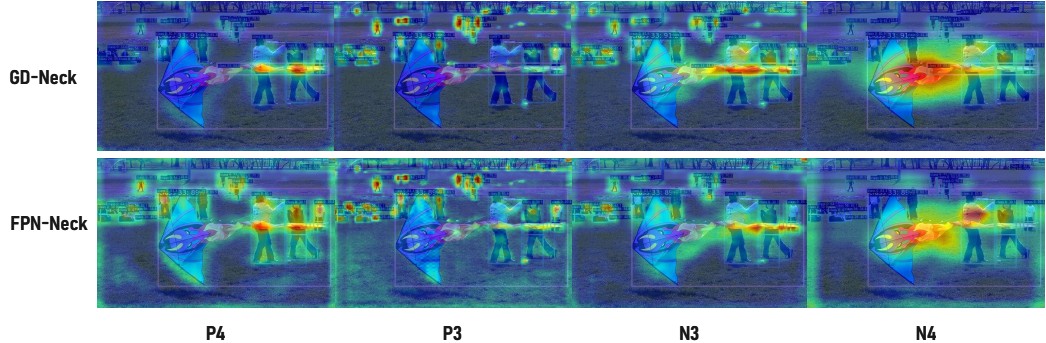

GD-Neck

FPN-Neck

P4          P3          N3          N4

Figure 7: Neck CAM visualization. We can observe that features at different levels exhibit distinct preferences for objects of different sizes. In the traditional grid-structured FPN, with increasing network depth and information interaction between different levels, the sensitivity of the feature map to object positions gradually diminishes, accompanied by information loss. Our proposed GD mechanism performs global fusion separately for high-level and low-level information, resulting in globally fused features that contain abundant position information for objects of various sizes.

## E  Discussion

### E.1  The differences between feature alignment module of Gold-YOLO and other similar works.

Both M2Det and RHF-Net have incorporated additional information fusion modules within their alignment modules. In M2Det, the SFAM module includes an SE block, while in RHF-Net, the Spatial Pyramid Pooling block is augmented with a Bottleneck layer. In contrast to M2Det and RHF-Net, Gold-YOLO leans towards functional separation among modules, segregating feature alignment and feature fusion into distinct modules. Specifically, the FAM module in GoldD-YOLO focuses solely on feature alignment. This ensures computational efficiency within the FAM module. And the LAF module efficiently merges features from various levels with minimal computational cost, leaving a greater portion of fusion and injection functionalities to other modules.

Based on the GD mechanism, achieving SOTA performance for the YOLO model can be accomplished using simple and easily accessible operators. This strongly demonstrates the effectiveness of the approach we propose. Additionally, during the network construction process, we intentionally opted for simple and thoroughly validated structures. This choice serves to prevent potential development and performance issues arising from certain operators being unsupported by deployment devices. As a result, it guarantees the usability and portability of the entire mechanism.

### E.2  Simple calculation operation

In the process of network construction, we have drawn from and built upon the experiences of previous researchers. Rather than focusing solely on performance improvements achieved by enhancing specific operators or local structures, our emphasis lies on the conceptual shift brought about by the GD mechanism in comparison to the traditional FPN structure. Through the GD mechanism, achieving SOTA performance for the YOLO model is attainable using simple and easily applicable operators. This serves as strong evidence for the effectiveness of the proposed approach.

Additionally, during the network construction process, we intentionally opted for simple and thoroughly validated structures. This choice serves to prevent potential development and performance issues arising from certain operators being unsupported by deployment devices. As a result, it guarantees the usability and portability of the entire mechanism. Moreover, this decision also creates opportunities for future performance enhancements.

The GD mechanism is a general concept and can be applied beyond YOLOs. We have extend GD mechanism to other models and obtain significant improvement. The experiments demonstrate

that our proposed GD mechanism exhibits robust adaptability and generalization. This mechanism consistently brings about performance improvements across different tasks and models.

