# OpenReview forum: "Gold-YOLO: Efficient Object Detector via Gather-and-Distribute Mechanism"
_NeurIPS.cc/2023/Conference — NeurIPS 2023 poster_

### Official Review · Reviewer_3APV · 2023-07-03

**Soundness:** 3 good
**Presentation:** 3 good
**Contribution:** 3 good
**Rating:** 6
**Confidence:** 4

**Summary:**

In this paper, the authors proposed a Gather-and-Distribute mechanism (GD) for efficient information exchange in YOLOs by globally fusing multi-level features and injecting the global information into higher levels. The proposed GD-YOLO architectures show good results compared with the existing YOLO series. The authors also presented a pre-training method, where we pre-train the backbone on ImageNet 1K using the MAE method, which improves the convergence speed and accuracy of the model.

**Strengths:**

1. In this paper, the authors proposed a gather-and-distribute mechanism to replace the traditional FPN structure. By using this unified module to gather and fuse information from all levels and subsequently distribute it to different levels, they can avoid the loss of information inherent in the traditional FPN structure and also enhance the neck’s partial information fusion capabilities without significantly increasing latency.
2. The paper is clear and organized, and easy to follow.
3. Authors have provided a comprehensive comparison between the proposed model and YOLOs.

**Weaknesses:**

1. The motivation is a little bit unclear. The advantage of this GD mechanism compared with traditional FPN are not very clear.
2. Based on the results, the tradeoff between accuracy and latency is pretty similar compared with baselines (YOLOv8).

**Questions:**

If the best trade-off between latency and accuracy is the main goal of this method, isn't fairer to compare with YOLOv8 instead of YOLOv6, since YOLOv8 has the closest latency with GDYolo?

**Limitations:**

1. The idea is pretty interesting but the advantage of this proposed method doesn't show a very convincing result.

---

> ### Author Rebuttal · Authors · 2023-08-09
>
> ### W-1: The motivation is a little bit unclear. The advantage of this GD mechanism compared with traditional FPN are not very clear.
>
> Thank you for your suggestions. In traditional concepts, features at different levels contain positional information of different-sized objects. Larger features encompass more low-dimensional texture information and the positions of smaller objects, while smaller features contain more high-dimensional information and the positions of larger objects. The original motivation behind traditional Feature Pyramid Networks (FPN) is that although features at different levels contain different information, these pieces of information can assist each other and enhance the network's performance. Many previous works have addressed the issue of information loss in interactions between different levels. However, due to the excessive number of paths and indirect interaction methods in the network, the previous FPN-based fusion structures still have drawbacks in low speed, cross-level information exchange and information loss.
>
> Here we focus on the efficiency of information interaction fusion and the integrity of information retention. Beyond FPN-based fusion structures, we propose a new scheme, i.e., gather-and-distribute mechanism which is new information fusion structure for detection, not a revised version of FPN. The visualized feature maps of different-level Class Activation Maps (CAM) of Figure 3 can be found in the global response PDF.
>
> We can observe that features at different levels exhibit distinct preferences for objects of different sizes. In the traditional grid-structured FPN, with increasing network depth and information interaction between different levels, the sensitivity of the feature map to object positions gradually diminishes, accompanied by information loss. Our proposed GD mechanism performs global fusion separately for high-level and low-level information, resulting in globally fused features that contain abundant position information for objects of various sizes. When injected into different branches, this not only enriches the information in each branch but also avoids information loss caused by "recursive" interaction.
>
>
>
> ### W-2: Based on the results, the tradeoff between accuracy and latency is pretty similar compared with baselines (YOLOv8).
>
> Thank you for your question. Our model shows significant performance improvements compared to previous works in the sizes of N/S/M (nano, small, medium), including YOLOv8. For YOLO-L, our GD-YOLO-L also exhibits performance improvements of +0/+9FPS and +0.6 AP/+1.1 AP compared to YOLOv6-3.0 and YOLOv8, respectively.
>
>
>
> ### Q-1: If the best trade-off between latency and accuracy is the main goal of this method, isn't fairer to compare with YOLOv8 instead of YOLOv6, since YOLOv8 has the closest latency with GDYolo?
>
> Thank you for your question. It's perfectly normal to have such confusion due to the naming inconsistencies in the YOLOv6 series, which has led to misunderstandings for many. The chronological order of model releases is as follows: YOLOv6, YOLOv7, YOLOv6-2.0, YOLOv8, YOLOv6-3.0. YOLOv6-3.0 was released after YOLOv8 and significantly improved model accuracy, allowing YOLOv6-3.0 to surpass YOLOv8 on the precision-speed curve. This result is also evident in our precision-speed curve graph.
>
>
>
> ### L-1: The idea is pretty interesting but the advantage of this proposed method doesn't show a very convincing result.
>
> Thank you for your question. Our core contribution lies in the proposal of the Gather-and-Distribute Mechanism (GD mechanism),  which is fast and effective.  Our method gathers the global information, and distribute this information into differtnt levels.  The GD mechanism is performed in both high resolution and low resolution to fully encourage information exchange.  By separately considering low-dimensional and high-dimensional information, we have constructed a global information fusion mechanism, thereby unifying the approaches for promoting information flow between different levels feature that were previously disparate.  Our proposed GD mechanism represents a more comprehensive improvement over previous SOTA detectors.
>
> The GD mechanism is a general concept and can be applied beyond YOLOs.  We have extend GD mechanism to other models and obtain significant improvement as show in the Experiment-1, you can find in the Global Author Rebuttal.  The aforementioned experiments demonstrate that our proposed GD mechanism exhibits robust adaptability and generalization.  This mechanism consistently brings about performance improvements across different tasks and models.

---

> ### Author Response · Authors · 2023-08-21
> **End of the discussion window approaching**
>
> Dear anonymous reviewers,
>
> Thank you for your constructive comments and valuable suggestions to improve this paper. ​If you have any more questions, we would be glad to discuss them with you.
>
> Thank you very much.
>
> Best regards, Author

---

### Official Review · Reviewer_bFVa · 2023-07-07

**Soundness:** 2 fair
**Presentation:** 2 fair
**Contribution:** 3 good
**Rating:** 6
**Confidence:** 3

**Summary:**

In this research, the authors propose a novel Gather-and-Distribute (GD) mechanism implemented through convolution and self-attention operations. This mechanism, incorporated into the GD-YOLO model, significantly enhances multi-scale feature fusion capabilities and achieves a remarkable balance between latency and accuracy across all model scales. Notably, the researchers introduce MAE-style pretraining to the YOLO-series models for the first time, enabling unsupervised pretraining benefits. The GD-YOLO-N variant achieves exceptional results, with a 39.9% Average Precision (AP) on the COCO val2017 dataset and a remarkable 1030 Frames Per Second (FPS) on a T4 GPU. These results surpass the previous state-of-the-art model, YOLOv6-3.0-N, with a similar FPS by 2.4%.

**Strengths:**

GD-YOLO is a quite impressive work, I think as it globalized the features from local and can retrieve features without tagging other layers feature with it.

**Weaknesses:**

1. While GD-YOLO-N cannot use the proposed backbone for limited capacity, what will be the approaches for mobile deployment as your main objective is YOLO for mobile deployment?
2. GD-YOLO-N implementation is not quite understandable while it misses a few parts like MIM and seems pretty same as YOLOv6 with EWA and AAT as stated.
3. Comparison with the Transformer based model could add a bit more value to this paper.
4. Abbreviations should be added after first introduction and then using the abbreviation is the ideal practice. For instance, in line 46, it should have been “... use Feature Pyramid Network (FPN) and…” instead of just FPN.
5. Line 45 to 54: This para basically represents the contribution part, which can be presented in bullet terms.
6. Line 58: “...SOTA YOLOv6…” : As per my knowledge, at this point, YOLOv8 is the SOTA. I would request for a checkup and update of this from your end.
7. Line 74-75: What strengths YOLOv8 takes from their predecessor, it is not mentioned.
8. Line 94: “In this study, we will…..” - it is recommended to use present tense, instead of future tense.
9. Table 1: Best values should be bolded.

Minor:
1. Line 32: “...detectors.Despite…, Line 138: “...alignment module(Low-FAM)...” - Spacing issue.
2. Line 257: Double comma issue.


**Questions:**

1. Why need to use LAMB Optimizer while already using SGD and manual scheduled Learning rate?
2. If you use EMA, then why again, LAMB is needed?


**Limitations:**

Limitations of GD-YOLO not added; however, I would suggest adding it.

---

> ### Author Rebuttal · Authors · 2023-08-09
>
> ### W-1: While GD-YOLO-N cannot use the proposed backbone for limited capacity, what will be the approaches for mobile deployment as your main objective is YOLO for mobile deployment?
>
> Thank you for your question. The main contribution of our work is the Gather-and-Distribute mechanism. Regarding the backbone, we are the first to pre-train the convolutional MIM in the YOLO series, aiming to explore the effect of MIM on the CNN backbone for real-time object detection task. We also compared the effects of MIM pre-training with ImageNet classification supervised pre-training on GD-YOLO-S in the appendix, and found a certain improvement. The reason for not implementing pre-training on GD-YOLO-N is that it is the most lightweight network within GD-YOLO. As such, its backbone struggles to show improvement in the MIM task.
>
> For mobile deployment, we have demonstrated strong support on ONNX and TensorRT, which clears the path for deploying our model on mobile devices.
>
> ### W-2: GD-YOLO-N implementation is not quite understandable while it misses a few parts like MIM and seems pretty same as YOLOv6 with EWA and AAT as stated.
>
> We understand your concerns about the GD-YOLO-N implementation. As you correctly noted, GD-YOLO-N does not incorporate the Mutual Information Maximization (MIM) pre-training, unlike our other models such as GD-YOLO-S. This decision was made because GD-YOLO-N is designed to be a lightweight model. Its backbone, due to its simplicity and size, does not benefit substantially from MIM pre-training.
>
> GD-YOLO-N might appear similar to YOLOv6 with EWA and AAT because these are common techniques used to improve performance in object detection tasks. However, our work introduces the Gather-and-Distribute mechanism, which we believe adds a unique value to our model. This mechanism, in conjunction with our other design choices, aims to strike a balance between model complexity and performance, particularly for deployment on mobile devices with limited computational resources.
>
> I hope this clarifies your concerns about GD-YOLO-N. Please feel free to ask if you have further questions.
>
> ### W-3: Comparison with the Transformer based model could add a bit more value to this paper.
>
> Thank you for your question. We have added a chart of object detection model based on Transformer .
>
> | model                | #Params | GFLOPs | FPS bs=1 | APval | Apval-50 |
> | -------------------- | ------- | ------ | -------- | ----- | -------- |
> | RT-DETR-L            | 32      | 110    | 114      | 53    | 71.6     |
> | RT-DETR-X            | 67      | 234    | 74       | 54.8  | 73.1     |
> | DINO-Deformable-DETR | 47      | 279    | 5        | 50.9  | 69       |
> | GD-YOLO-N            | 5.6     | 12.1   | 563      | 39.9  | 55.9     |
> | GD-YOLO-S            | 21.5    | 46     | 286      | 46.4  | 63.4     |
> | GD-YOLO-M            | 41.3    | 87.5   | 152      | 51.1  | 68.5     |
> | GD-YOLO-L            | 75.1    | 151.7  | 88       | 53.3  | 70.9     |
>
> This highlights the absolute performance advantage of GD-YOLO in the domain of small models . Despite RT-DETR significant advantages over YOLO-L/X, due to the structural constraints of the transformer, RT-DETR cannot be further downsized. This makes GD-YOLO still the best choice in the field of small models.
>
> ### W-6: Line 58: “...SOTA YOLOv6…” : As per my knowledge, at this point, YOLOv8 is the SOTA. I would request for a checkup and update of this from your end.
>
> Thank you for your question. It's perfectly normal to have such confusion due to the naming inconsistencies in the YOLOv6 series, which has led to misunderstandings for many. The chronological order of model releases is as follows: YOLOv6, YOLOv7, YOLOv6-2.0, YOLOv8, YOLOv6-3.0. YOLOv6-3.0 was released after YOLOv8 and significantly improved model accuracy, allowing YOLOv6-3.0 to surpass YOLOv8 on the precision-speed curve. This result is also evident in our precision-speed curve graph.
>
> ### W-7: Line 74-75: What strengths YOLOv8 takes from their predecessor, it is not mentioned.
>
> Thank you for your question. YOLOv8 introduces a new Conv block called C2f, which compared to YOLOv5's C3 block, enhances the gradient path even further. In terms of loss calculation, YOLOv8 adopts the TaskAlignedAssigner positive sample assignment strategy and introduces the Distribution Focal Loss. YOLOv8 is more of an integration of design elements from various YOLO series models such as YOLOX, YOLOv6, and YOLOv7, and leans more towards engineering.
>
>
>
> ### Format problem：
>
> - W-4: Abbreviations should be added after first introduction and then using the abbreviation is the ideal practice. For instance, in line 46, it should have been “... use Feature Pyramid Network (FPN) and…” instead of just FPN.
> - W-8: Line 94: “In this study, we will…..” - it is recommended to use present tense, instead of future tense.
> - W-9: Table 1: Best values should be bolded.
> - Minor:
>   1. Line 32: “...detectors.Despite…, Line 138: “...alignment module(Low-FAM)...” - Spacing issue.
>   2. Line 257: Double comma issue.
>
> Thanks for your suggestions, we will correct these formatting and grammar problems in the revision.
>
>
>
> ### Q-1: Why need to use LAMB Optimizer while already using SGD and manual scheduled Learning rate?
>
> Thank you for your question. We utilize the LAMB optimizer only for MIM pre-trained backbones, following the practices in SparK. For the rest of the training process and self-distillation, we adopt SGD.
>
> ### Q-2: If you use EMA, then why again, LAMB is needed?
>
> Thank you for your question. As with Q-1, we only followed SparK's design in pre-training, using the LAMB optimizer.
>
> ### Limitations: Limitations of GD-YOLO not added; however, I would suggest adding it.
>
> Thank you for your question. Limitations can be found in section C Broader impacts and limitations in the Appendix.

---

> ### Author Response · Authors · 2023-08-21
> **End of the discussion window approaching**
>
> Dear anonymous reviewers,
>
> Thank you for your constructive comments and valuable suggestions to improve this paper. ​If you have any more questions, we would be glad to discuss them with you.
>
> Thank you very much.
>
> Best regards, Author

---

### Official Review · Reviewer_c8e6 · 2023-07-09

**Soundness:** 3 good
**Presentation:** 3 good
**Contribution:** 2 fair
**Rating:** 5
**Confidence:** 4

**Summary:**

This paper studies the problem of efficient object detector and proposes the Gather-and-Distribute mechanism (GD) mechanism to alleviate the information fusion problem. The experiments on the COCO dataset demonstrate the effectiveness of the proposed method.

**Strengths:**

+ This paper studies an important topic, efficient object detection, and achieves a great balance between accuracy and speed in its results.
+ The ablation studies have been provided to verify the effectiveness of the proposed module.

**Weaknesses:**

- The structural design of this paper is quite confusing in some aspects, such as the choice of where to inject information. For example, in Low-GD, semantic information is only injected into P3 and P4, while one would expect that global semantic information could also benefit the P5 branch. Similarly, in High-GD, information is only injected into N4 and N5. The authors should provide the rationale and advantages of this design choice to address these doubts.

- In line #157, the authors mention they were inspired by [28]. However, the injection design is more closely related to Topformer, which bears greater relevance to the structure of the current paper. The authors should consider discussing and citing Topformer as another source of inspiration or relevant related work.

[a] TopFormer: Token Pyramid Transformer for Mobile Semantic Segmentation, CVPR 2022.

**Questions:**

Please refer to the Weaknesses.

---

> ### Author Rebuttal · Authors · 2023-08-09
>
> ### W-1: The structural design of this paper is quite confusing in some aspects, such as the choice of where to inject information. For example, in Low-GD, semantic information is only injected into P3 and P4, while one would expect that global semantic information could also benefit the P5 branch. Similarly, in High-GD, information is only injected into N4 and N5. The authors should provide the rationale and advantages of this design choice to address these doubts.
>
> Thank you for your question. When designing the fusion mechanism, we aim for the two branches in the network to focus on low-dimensional and high-dimensional information respectively. Following this principle, we have designed and implemented two branches: Low-GD and High-GD. This design approach results in the fused features from the two branches having richer low-dimensional and high-dimensional information respectively.
>
> Additionally, it is commonly believed that features at different levels contain positional information of objects of varying sizes. For instance, larger features encompass more low-dimensional texture information and positions of smaller objects, while smaller features contain more high-dimensional structural information and positions of larger objects. This aligns well with the characteristics of the two branches in the GD mechanism. Therefore, the global information generated by Low-GD is injected into feature maps with larger feature sizes, whereas the global information generated by High-GD is injected into feature maps with smaller feature sizes.
>
> Such a choice is also based on a trade-off between accuracy and speed. We conducted ablation experiments to validate this idea:
>
> | model     | Inject level          | AP          | AP-50       | FPS-32 | Params | FLOPs   |
> | --------- | --------------------- | ----------- | ----------- | ------ | ------ | ------- |
> | GD-YOLO-S | [P3/P4]-[N4/N5]       | 45.4 / 46.1 | 62.5 / 63.3 | 446    | 21.5 M | 46.00 G |
> | GD-YOLO-S | [P3/P4/P5]-[N3/N4/N5] | 46.3 / 46.6 | 63.4 / 64.0 | 397.76 | 23.2 M | 50.43 G |
>
> The experimental results indicate that as the inject level increases, the model's accuracy also increases. However, at the same time, the speed of the model decreases. The final version provided in the paper represents our optimal choice in balancing speed and accuracy.
>
>
>
>
>
> ### W-2:  In line #157, the authors mention they were inspired by [28]. However, the injection design is more closely related to Topformer, which bears greater relevance to the structure of the current paper. The authors should consider discussing and citing Topformer as another source of inspiration or relevant related work.
>
> Thank you for your reminder. Topformer and Seaformer are both important sources of inspiration for us. We’ll include the discussion with M2Det [1] and RHF-Net [2] in our final version:
>
> Built upon the concept of global information fusion, TopFormer has achieved remarkable results in semantic segmentation tasks. Expanding on the foundation of TopFormer, we have taken a step further by separately considering high-dimensional and low-dimensional information. We have meticulously designed two branches, namely Low-GD and High-GD, infusing the notion of global information fusion into the realm of object detection. As a result, we have achieved SOTA performance at object detection task.

---

> ### Author Response · Authors · 2023-08-21
> **End of the discussion window approaching**
>
> Dear anonymous reviewers,
>
> Thank you for your constructive comments and valuable suggestions to improve this paper. ​If you have any more questions, we would be glad to discuss them with you.
>
> Thank you very much.
>
> Best regards, Author

---

### Official Review · Reviewer_YJPg · 2023-07-10

**Soundness:** 2 fair
**Presentation:** 2 fair
**Contribution:** 3 good
**Rating:** 5
**Confidence:** 3

**Summary:**

In this paper, the authors proposed an efficient object detection network to make a new trade-off between efficiency and effectivity. In the framework, a lightweight adjacent-layer fusion module termed as gather-and-distribute (GD) mechanism is proposed to take place the conventional neck module in general detectors. Experimental results on COCO built-on the YOLO-series off-the-peg detectors demonstrate the effectiveness of the proposed method.

**Strengths:**

-The bilateral interaction of information in  multi-layer layers is interesting, and I like the work that simple changes can bring significant improvement for foundational tasks.

-The experimental setup of ablation studies on COCO is helpful for follow-up work in the future.

**Weaknesses:**

The main weakness of this paper is the limited technical novelty and the relatively inadequate experiments. First, although the authors try to explain that the proposed neck module of the GD mechanism can improve the detection accuracy of the detection task and benefit the computational efficiency, the novelty of the proposed module can be seen as a compromise similar to the stack of the existing technologies. Additionally, the experiment is carried out only on YOLO-series methods, and there is a lack of testing and verification on other datasets.

This paper needs further modification in terms of layout design, e.g., the fonts in the Fig.4 are generally small and unclear, which is very difficult for review.

Page 3 Line 82-83, Improving ->can improve, introduce -> introduced.

**Questions:**

The innovation of the proposed method needs to be improved comprehensively, and there is a lack of more substantial analysis and understanding in terms of innovation in technology. Then, it is necessary to verify the scalability and robustness of the proposed method in more experiments and even more tasks. In addition, the writing and layout concept of the article still needs to be improved.

**Limitations:**

Applicable

---

> ### Author Rebuttal · Authors · 2023-08-09
>
> ### W1: The novelty of the proposed module can be seen as a compromise similar to the stack of the existing technologies, and lack of testing and verification on other datasets.
>
> Thank you for your suggestions. Our core contribution lies in the proposal of the Gather-and-Distribute Mechanism (GD mechanism),  By separately considering low-dimensional and high-dimensional information, we have constructed a global information fusion mechanism, thereby unifying the approaches for promoting information flow between different levels feature that were previously disparate. Our proposed GD mechanism represents a more comprehensive improvement over previous works.
>
> In the process of network construction, we have drawn from and built upon the experiences of previous researchers. Rather than focusing solely on performance improvements achieved by enhancing specific operators or local structures, our emphasis lies on the conceptual shift brought about by the GD mechanism in comparison to the traditional FPN structure. Through the GD mechanism, achieving SOTA performance for the YOLO model is attainable using simple and easily applicable operators. This serves as strong evidence for the effectiveness of the proposed approach.
>
> Additionally, during the network construction process, we intentionally opted for simple and thoroughly validated structures. This choice serves to prevent potential development and performance issues arising from certain operators being unsupported by deployment devices. As a result, it guarantees the usability and portability of the entire mechanism. Moreover, this decision also creates opportunities for future performance enhancements.
>
> The GD mechanism is a general concept and can be applied beyond YOLOs. We have extend GD mechanism to other models and obtain significant improvement as show in the Experiment-1, you can find in the Global Author Rebuttal. The aforementioned experiments demonstrate that our proposed GD mechanism exhibits robust adaptability and generalization. This mechanism consistently brings about performance improvements across different tasks and models.
>
>
> ### Format problem
>
> - W-2: Fig.4 are generally small and unclear
> - W-3: Page 3 Line 82-83, Improving ->can improve, introduce -> introduced.
>
> Thanks for your suggestions, we will correct these formatting and grammar problems in the revision.
>
>
> ### Q-1: Lack of in-depth analysis and understanding regarding innovation, along with insufficient evidence of the scalability and robustness of the proposed method.
>
> Thank you for your suggestions. We’ll include more discussion with related works on multi-scale features in the final version. The specific content please refer to the Discuss-1 in global  response.
>
> In order to demonstrate the scalability and robustness of our proposed GD mechanism, in addition to the experiments conducted in semantic segmentation and instance segmentation tasks as mentioned in W1, we have conducted the following additional experiments to supplement our idea:
>
> | model| AP| AP-50|FPS-32|Params|FLOPs|
> |-|-|-|-|-|-|
> | GD-YOLO-S|45.4 / 46.1| 62.5 / 63.3| 446| 21.5 M | 46.00 G |
> | GD-YOLO-S-all_trans  | 46.9 / 47.0 | 64.1 / 64.4 | 258.56 | 25.2 M | 57.95 G|
> | GD-YOLO-S-all_conv| 45.6 / 46.1 | 62.5 / 63.2 | 417.76 |26.6 M| 47.06 G|
> | GD-YOLO-S-inject_all| 46.3 / 46.6 | 63.4 / 64.0 | 397.76 |23.2 M| 50.43 G|
>
> We have developed three variants based on GD-YOLO-S:
>
> 1. GD-YOLO-S-all_trans: Replaces the fusion operators in both Low-IFM and High-IFM with Transformers.
> 2. GD-YOLO-S-all_conv: Replaces the fusion operators in both Low-IFM and High-IFM with Convolutions.
> 3. GD-YOLO-S-inject_all: Adds Inject models to all P-level and N-level, injecting global information into each level.
>
> From the results of experiments 1 and 2, it is evident that the model's performance does not significantly degrade with changes in the operators within the fusion module. This demonstrates the strong robustness of the GD mechanism and its insensitivity to operators. The GD mechanism can be effortlessly combined with any operators, introducing new features by incorporating new operators.
>
> In experiment 3, we extended the number of layers for information injection within the GD mechanism, resulting in improved accuracy at the cost of reduced speed. Additionally, in section 4.3.1 "Ablation study on GD structure" of the main text, we conducted detailed ablation experiments on various modules, further demonstrating the scalability of the proposed GD mechanism. In practical applications or different tasks, these modules can be freely combined and adjusted based on specific requirements to achieve optimal performance.
>
> As per your suggestions, we will include a more in-depth analysis of multi-scale features in the revised version. And correct the grammar problems in the article, improve the graphic layout.

---

> ### Author Response · Authors · 2023-08-20
> **End of the discussion window approaching**
>
> Dear anonymous reviewers,
>
> Thank you for your constructive comments and valuable suggestions to improve this paper. ​If you have any more questions, we would be glad to discuss them with you.
>
> Thank you very much.
>
> Best regards, Author

---

### Official Review · Reviewer_CoJm · 2023-08-02

**Soundness:** 3 good
**Presentation:** 4 excellent
**Contribution:** 2 fair
**Rating:** 5
**Confidence:** 5

**Summary:**

The paper presents a real-time object detection method for the YOLO series, introducing a 'Gather-and-Distribute' (GD) mechanism. Despite achieving good results on the COCO dataset, the paper lacks significant novelty and doesn't significantly advance multi-scale feature fusion or FPN-based methods. A deeper comparison with prior work could strengthen its scientific contribution.

**Strengths:**

1. The paper exhibits commendable clarity with well-written content and lucidly presented figures, making it easy to understand.

2. Performance-wise, the proposed contribution demonstrates impressive results on the COCO dataset in terms of both accuracy and computational efficiency.

3. The introduction of the 'Gather-and-Distribute' (GD) mechanism is particularly striking. This method enhances multi-scale feature fusion capabilities, striking an excellent balance between latency and accuracy across different model scales.

**Weaknesses:**

Weaknesses:

1. Despite achieving strong results on the COCO dataset in terms of accuracy and efficiency, the paper lacks substantial scientific novelty. The method, while technically sound, doesn't significantly advance the field.

2. The manuscript's focus is on real-time object detection and multi-scale features for the YOLO-Series. However, the related work section needs to delve more into multi-scale features.

3. The concepts of Low/High-FAM and the lightweight adjacent layer fusion (LAF) module are not new in the field, having been discussed in M2Det [1] and [2] respectively.

Suggestions:

1. The authors need to restructure the related work section to better represent and compare with multi-scale features[1] or FPN-based methods [2-8].

2. Adding more comparative analyses and surveys related to multi-scale features and FPN-based methods will establish their work as more than a minor modification of previous works.

3. Despite YOLO-Series being known for speed and efficiency, GD-YOLO appears to be slower than the baseline (YOLOV6: v3). The authors should address this discrepancy.

4. To reiterate, the work seems to be more application-focused with minimal contribution to the field. I recommend the authors address the points above to enhance their scientific contribution, which may change the review score. Otherwise, the work might not meet the standards of a top-tier conference.


[1] Zhao, Q., Sheng, T., Wang, Y., Tang, Z., Chen, Y., Cai, L., & Ling, H. (2019, July). M2det: A single-shot object detector based on multi-level feature pyramid network. In Proceedings of the AAAI conference on artificial intelligence (Vol. 33, No. 01, pp. 9259-9266).

[2] Chen, P. Y., Hsieh, J. W., Wang, C. Y., & Liao, H. Y. M. (2020). Recursive hybrid fusion pyramid network for real-time small object detection on embedded devices. In Proceedings of the IEEE/CVF Conference on Computer Vision and Pattern Recognition Workshops (pp. 402-403).

[3] Quan, Y., Zhang, D., Zhang, L., & Tang, J. (2023). Centralized feature pyramid for object detection. IEEE Transactions on Image Processing.

[4]Yang, G., Lei, J., Zhu, Z., Cheng, S., Feng, Z., & Liang, R. (2023). AFPN: Asymptotic Feature Pyramid Network for Object Detection. arXiv preprint arXiv:2306.15988.

[5] Jin, Z., Yu, D., Song, L., Yuan, Z., & Yu, L. (2022, October). You should look at all objects. In European Conference on Computer Vision (pp. 332-349). Cham: Springer Nature Switzerland.

[6] Chen, Q., Wang, Y., Yang, T., Zhang, X., Cheng, J., & Sun, J. (2021). You only look one-level feature. In Proceedings of the IEEE/CVF conference on computer vision and pattern recognition (pp. 13039-13048).

[7] Jin, Z., Liu, B., Chu, Q., & Yu, N. (2020). SAFNet: A semi-anchor-free network with enhanced feature pyramid for object detection. IEEE Transactions on Image Processing, 29, 9445-9457.

[8] Chen, P. Y., Chang, M. C., Hsieh, J. W., & Chen, Y. S. (2021). Parallel residual bi-fusion feature pyramid network for accurate single-shot object detection. IEEE Transactions on Image Processing, 30, 9099-9111.

**Questions:**

1. Could you elaborate on why YOLOv7-E6E has been chosen as the "L" model for comparison, given that its size is 1280? It seems that YOLOv7-X would be a more suitable choice for a fair comparison.

2. The relevance of the section "Masked image modeling pre-training" isn't clear, as it doesn't appear to directly relate to multi-scale features and FPN-based methods. Additionally, the results from this section don't seem to add significant value. Would it be more beneficial to include additional comparisons or ablation studies for multi-scale features in lieu of this section?

**Limitations:**

I want to highlight the author's commendable acknowledgment of the potential military applications of their model. They clearly state their commitment to prevent such uses. This level of social impact consideration is laudable and sets a positive precedent for responsible research conduct.

---

> ### Author Rebuttal · Authors · 2023-08-09
>
> ## Weaknesses
>
> ### W-1: The paper lacks substantial scientific novelty, and doesn't significantly advance the field.
>
> Thank you for your suggestions. Our core contribution lies in the proposal of the Gather-and-Distribute Mechanism (GD mechanism), which is fast and effective. Our method gathers the global information, and distribute this information into differtnt levels. The GD mechanism is performed in both high dimension and low dimension to fully enhance information exchange. By separately considering low-dimensional and high-dimensional information, we have constructed a global information fusion mechanism, thereby unifying the approaches for promoting information flow between different levels feature that were previously disparate. Our proposed GD mechanism represents a more comprehensive improvement over previous SOTA detectors.
>
> The GD mechanism is a general concept and can be applied beyond YOLOs. We have extend GD mechanism to other models and obtain significant improvement as show in the Experiment-1, you can find in global response.
>
> ### W-2: The related work section needs to delve more into multi-scale features.
>
> Thank you for your suggestions. We’ll include more discussion with related works on multi-scale features in the final version. The specific content please refer to the Discuss-1 in global response.
>
> ### W-3: The issue of similarity between the FAM/LAF modules and M2Det [1] / [2].
>
> Thank you for raising the question.  Rather than focusing on improving specific operators or local structures, we propose a new Gather-and-Distribute (GD) mechanism compared to the traditional FPN structure, without relying on the characteristics of special operators.
>
> We’ll include the discussion with M2Det [1] and RHF-Net [2] in our final version. The specific content please refer to the Discuss-2 in global response.
>
> ## Suggestions
>
> ### S-1: The authors need to restructure the related work section to better represent and compare with multi-scale features[1] or FPN-based methods [2-8].
>
> Thank you for your suggestions. We will follow your advice and incorporate a more in-depth investigation and analysis of multi-scale features in the revised version. Additionally, we will cite [1-8] and other relevant articles to comprehensively outline the historical development of the relevant field.
>
> We’ll include more discussion with related works on multi-scale features in the final version. The specific content please refer to the Discuss-1 in global response.
>
> In addition, we tested the inference speed of some of the open source models on the V100 and plotted the accuracy-speed curve. It proves the excellent performance of our model. The experimental results can be found in Experiment-2. And the Figure 2 can be found in the global response PDF.
>
> ### S-2: Adding more comparative analyses and surveys related to multi-scale features and FPN-based methods will establish their work as more than a minor modification of previous works.
>
> Thank you for your suggestions. The specific content please refer to the Discuss-1 in global response.
>
> ### S-3: GD-YOLO appears to be slower than the baseline (YOLOV6: v3)
>
> Thank you for your feedback. In the comparison with previous generations of YOLO models, our focus lies more on the trade-off between speed and accuracy improvements, rather than solely aiming for the utmost speed or precision in a single model. Of course, we have developed a smaller and faster model (GD-YOLO-N-2), and the specific performance is detailed in the table below:
>
> | model|FPS-32|AP|AP:50|
> |-|-|-|-|
> |YOLOv6-3.0-N|1187|37.0 / 37.5|52.7 / 53.1|
> |GD-YOLO-N-2|1211|38.09 / 38.41|54.92 / 55.14 |
>
> GD-YOLO-N is built by reducing the parameter count in the neck section (removing the LAF module and High-GD branch), resulting in a smaller and faster model, GD-YOLO-N-2. This model outperforms the current SOTA YOLOv6-3.0-N in both accuracy and speed. The reason we present results for GD-YOLO-N in the article, rather than GD-YOLO-N-2, is to maintain consistent structures across models of different sizes and ensure result continuity.
>
> ### S-4: The work seems to be more application-focused with minimal contribution to the field.
>
> Thanks for your suggestion. Our contribution is not minimal since we propose a new scheme called GD mechanism, which is a general concept and can be applied beyond YOLOs (details can be found in the responses in W-1/2). Moreover, we think our paper is proper for NeurIPS and appeals to the research community, as the Call For Papers in NeurIPS23 stated “We invite submissions presenting new and original research on topics including but not limited to the following: Applications (e.g., vision, language, speech and audio)”.
>
> ## Questions
>
> ### Q-1: Why YOLOv7-E6E has been chosen as the "L" model for comparison
>
> Thank you for sharing the information. Below is the test result for YOLOv7-X whose performance is poorer than YOLOv7-E6E. A more intuitive  Figure 1 can be found in the global response PDF.
>
> |model|FPS-32|AP|AP:50|
> |-|-|-|-|
> |YOLOv7-X|73.92|52.9|71.2|
>
> ### Q-2: The value and relevance of pre-training
>
> Thank you for your question. We would like to emphasize that the use of MIM pre-training is not meant to have a direct correlation with Gather-and-Distribute methods. Instead, it's intended to act as a practical and robust technique in real-world applications, enhancing the overall performance of the model without adding to the inference latency. To the best of our knowledge, we are the pioneers in applying MIM pre-training to the YOLO series. The experimental results presented in the appendix demonstrate the advantages of pre-training with ImageNet classification.

---

> ### Author Response · Authors · 2023-08-20
> **End of the discussion window approaching**
>
> Dear anonymous reviewers,
>
> Thank you for your constructive comments and valuable suggestions to improve this paper. ​If you have any more questions, we would be glad to discuss them with you.
>
> Thank you very much.
>
> Best regards, Author

---

> ### Comment · Reviewer_CoJm · 2023-08-20
>
> I appreciate that the authors addressed all of my concerns. They've acknowledged potential shortcomings regarding scientific novelty and have effectively cited related work to provide context. Moreover, the exploration of the motivation behind GD, as well as comprehensive discussions on FPN and multi-scale features, enhance the paper's value. The additional experimental results shed more light on its contributions, especially concerning multi-scale features.
>
> Additionally, the authors have incorporated some of my suggestions and presented convincing results.
>
> However, I would still recommend removing the section titled "Masked image modeling pre-training." While this topic is interesting, it might be more fitting for a separate paper. For this work, focusing on the contributions of the multi-scale feature would be more appropriate.
>
> To summarize, I am satisfied with the authors' responses in the rebuttal. The comparison of various multi-scale feature methods, paired with thorough experimentation, bolsters the paper's scientific credibility. Given these considerations, I have adjusted my rating to "Borderline Accept."

---

> > ### Author Response · Authors · 2023-08-21
> >
> > Dear anonymous reviewers,
> >
> > We sincerely appreciate you taking time to review our responses and contributing to improve this paper. We will carefully follow reviewer's advice to incorporate the addressed points in updated version.
> >
> > Best regards, Author

---

### Author Rebuttal · Authors · 2023-08-09

### Experiment-1:

- Instance Segmentation Task

  Replace different Necks in Mask R-CNN and train/test on the COCO instance dataset.

  | model| Neck|FPS|Bbox mAP | Bbox mAP:50 | Segm mAP | Segm mAP:50 |
  |-|-|-|-|-|-|-|
  | MaskRCNN-ResNet50| FPN| 21.6|38.2| 58.8| 34.7| 55.7|
  | MaskRCNN-ResNet50| AFPN| 19.1|36.0|53.6| 31.8| 50.7|
  | MaskRCNN-ResNet50|PAFPN | 20.2|37.9|58.6| 34.5| 55.3|
  | MaskRCNN-ResNet50| GD| 18.7| 40.7|59.5| 36.0|56.4|

- Semantic Segmentation Task

  Replace the Neck in PointRend and train/test on the Cityscapes dataset.

  | model|Neck|FPS| mIoU| mAcc|aAcc|
  | - |-|-|-|-|-|
  | PointRend-ResNet50|FPN|11.21| 76.47| 84.05| 95.96|
  | PointRend-ResNet50| GD|11.07| 78.54| 85.60| 96.12|
  | pointrend-ResNet101| FPN|8.76| 78.3| 85.705|96.23|
  | PointRend-ResNet101| GD|11.07|80.01|86.15|96.34|

- Performance of GD mechanism on other object detection models

  Replace necks in efficientdet and train/test on the COCO instance dataset.

  | model|Neck|FPS|AP|
  |-|-|-|-|
  | EfficientDet|BiFPN| 6.0|34.4 |
  | EfficientDet|GD|5.7|38.8|

### Experiment-2:

- FPN-like model comparison

  | Model|FPS| AP-val2017 |
  |-|-|-|
  | M2Det| 15.178| 37.8|
  | AFPN-R50-640x640 | 28.1| 39|
  | AFPN-R50-800x1000 | 23.8| 41|
  | AFPN-R101-800x1000 | 16.5| 42.3|
  | YOLOF| 36.63| 37.7|
  | YOLOF-R101| 25.38| 39.8|
  | YOLOF-X101| 10.88| 42.3|
  | CFPNet-S| 331| 41.1|
  | CFPNet-M| 165| 46.4|
  | CFPNet-L| 95| 49.4|
  | GD-YOLO-N| 684| 39.9|
  | GD-YOLO-S| 337| 46.4|
  | GD-YOLO-M| 177| 51.1|
  | GD-YOLO-L| 110| 53.3|

### Discuss-1: A exploration of the motivation behind GD, as well as in-depth discussions on FPN and multi-scale features.

Traditionally, features at different levels carry positional information about objects of various sizes. Larger features encompass low-dimensional texture details and positions of smaller objects. In contrast, smaller features contain high-dimensional information and positions of larger objects. The original idea behind Feature Pyramid Networks (FPN) is that these diverse pieces of information can enhance network performance through mutual assistance. Previous works have addressed the information loss problem when interacting between different levels. For instance, M2Det[1] introduced an efficient MLFPN architecture with U-shape and Feature Fusion Modules. Ping-Yang Chen[2] improved interaction between deep and shallow layers using bidirectional fusion modules. Unlike these inter-layer works, [3] explored individual feature information using the Centralized Feature Pyramid (CFP) method. Additionally, [4] extended FPN with the Asymptotic Feature Pyramid Network (AFPN) to interact across non-adjacent layers. In response to FPN's limitations in detecting large objects, [5] proposed a refined FPN structure. YOLO-F [6] achieved state-of-the-art performance with single-level features. SAFNet [7] introduced Adaptive Feature Fusion and Self-Enhanced Modules. [8] presented a parallel FPN structure for object detection with bi-directional fusion.

However, due to the excessive number of paths and indirect interaction methods in the network, the previous FPN-based fusion structures still have drawbacks in low speed, cross-level information exchange and information loss.

Here we focus on the efficiency of information interaction fusion and the integrity of information retention. Beyond FPN-based fusion structures, we propose a new scheme, i.e., gather-and-distribute mechanism which is new information fusion structure for detection, not a revised version of FPN. As shown in the visualized feature maps of different-level Class Activation Maps (CAM) in the appendix.

We can observe that features at different levels exhibit distinct preferences for objects of different sizes. In the traditional grid-structured FPN, with increasing network depth and information interaction between different levels, the sensitivity of the feature map to object positions gradually diminishes, accompanied by information loss. Our proposed GD mechanism performs global fusion separately for high-level and low-level information, resulting in globally fused features that contain abundant position information for objects of various sizes. When injected into different branches, this not only enriches the information in each branch but also avoids information loss caused by "recursive" interaction.

### Discuss-2: Differences in the feature alignment module between GD-YOLO and other similar works

Both M2Det [1] and RHF-Net [1] have incorporated additional information fusion modules within their alignment modules. In M2Det, the SFAM module includes an SE block, while in RHF-Net, the Spatial Pyramid Pooling block is augmented with a Bottleneck layer. In contrast to M2Det and RHF-Net, GD-YOLO leans towards functional separation among modules, segregating feature alignment and feature fusion into distinct modules. Specifically, the FAM module in GD-YOLO focuses solely on feature alignment. This ensures computational efficiency within the FAM module. And the LAF module efficiently merges features from various levels with minimal computational cost, leaving a greater portion of fusion and injection functionalities to other modules.

Based on the GD mechanism, achieving SOTA performance for the YOLO model can be accomplished using simple and easily accessible operators. This strongly demonstrates the effectiveness of the approach we propose. Additionally, during the network construction process, we intentionally opted for simple and thoroughly validated structures. This choice serves to prevent potential development and performance issues arising from certain operators being unsupported by deployment devices. As a result, it guarantees the usability and portability of the entire mechanism.

---

> ### Comment · Area_Chair_z9qf · 2023-08-13
> **lets followup author response**
>
> Hi all,
>
> Thanks for serving as the reviewers for this submission. As the authors have already provided their responses. Now let's start further discussion. Here is a to-do list:
>
> (1) Please acknowledge the authors when you finish reading their responses.
>
> (2) Please indicate whether you have any further questions for the authors such that they can continue to response.
>
> (3) Please indicate whether you are willing to change the ratings.
>
> best,
> The AC

---

> ### Author Response · Authors · 2023-08-15
>
> Dear area chair and anonymous reviewers,
>
> Thank you for your constructive comments and valuable suggestions to improve this paper.      ​We conducted further experiments on new tasks and datasets, and subsequently engaged in additional discussions and explanations based on the outcomes of these experiments.     If you have any questions, we are glad to discuss them with you.
>
> Thank you very much.
>
> Best regards,
> Author

---

### Decision · Program_Chairs · 2023-09-21

**Decision:**

Accept (poster)

**Comment:**

This work proposes an efficient object detection method, named GD-YOLO. The authors introduce a Gather-and-Distribute mechanism to boost the multi-scale feature fusion capability. Extensive experiments show the great performance of this method. The reviewers acknowledge its outstanding performance but also point out the lack of novelty (by Reviewer CoJm and YJPg), and unclear motivation (by Reviewer c8e6 and 3APV). After the rebuttal, the concerns of Reviewer CoJm are addressed and he/she raises the rating to “Borderline Accept”. Considering its scores (2 Weak Accept, 2 Borderline accept, and 1 Borderline reject), I lean towards acceptance. I also suggest the authors revise the paper according to the reviewers' suggestions and further clarify the motivation of the “Gather-and-Distribute mechanism”.